# Mixbiotic society measures: Assessment of community well-going as living system

**Takeshi Kato**[1]*, **Jyunichi Miyakoshi**[2], **Tadayuki Matsumura**[2], **Ryuji Mine**[1], **Hiroyuki Mizuno**[2], **Yasuo Deguchi**[3]

**1** Hitachi Kyoto University Laboratory, Kyoto University, Kyoto, Japan, **2** Hitachi Kyoto University Laboratory, Center for Exploratory Research, Research & Development Group, Hitachi, Ltd., Tokyo, Japan, **3** Department of Philosophy, Graduate School of Letters, Kyoto University, Kyoto, Japan

* kato.takeshi.3u@kyoto-u.ac.jp

**Data Availability Statement:** All relevant data are within the manuscript.

**Funding:** This work was supported by JST RISTEX, Grant Number JPMJRS22J5, Japan. The funders had no role in study design, data collection and

## Abstract

Social isolation and fragmentation represent global challenges, with the former stemming from a lack of interaction and the latter from exclusive mobs—both rooted in communication issues. Addressing these challenges, the philosophical realm introduces the concept of the "mixbiotic society." In this framework, individuals with diverse freedoms and values mix together in physical proximity with diverse mingling, recognizing their respective "fundamental incapacities" and uniting in solidarity. This study aims to provide novel measures to balance freedom and solidarity, specifically the intermediate between isolation and mobbing, within a mixbiotic society. To achieve this, we introduce simplified measures to evaluate dynamic communication patterns. These measures complement traditional social network analysis of static structures and conventional entropy-based assessments of dynamic patterns. Our specific hypothesis posits that the measures corresponding to four distinct phases are established by representing communication patterns as multidimensional vectors. These measures include the mean of Euclidean distance to quantify "mobism" for fragmentation, the relative distance change for "atomism" indicating isolation, and a composite measure derived from multiplying the mean and variance of cosine similarity for "mixism," reflecting the well-going state of a mixbiotic society. Additionally, nearly negligible measures correspond to "nihilism." Through the evaluation of seven real-society datasets (high school, primary school, workplace, village, conference, online community, and email), we demonstrate the utility of the "mixism" measure in assessing freedom and solidarity in society. These measures can be employed to typify communities on a radar chart and a communication trajectory graph. The superiority of the measures lies in their ability to evaluate dynamic patterns, ease of calculation, and easily interpretable meanings compared to conventional analyses. As a future development, alongside additional validation using diverse datasets, the mixbiotic society measures will be employed to analyze social issues and applied in the fields of digital democracy and platform cooperativism.

analysis, decision to publish, or preparation of the manuscript.

**Competing interests:** The authors have declared that no competing interests exist.

## Introduction

Social isolation and fragmentation are global issues that have serious consequences. Social isolation, defined as an objective lack of interactions with others or the wider community, and loneliness, characterized by the subjective feeling of the absence of a social network or companion, are well-documented phenomena [1]. Isolation and loneliness have detrimental effects on health and well-being, necessitating targeted interventions to engage with affected populations [1, 2]. Conversely, social fragmentation refers to the absence or underdevelopment of connections between a society and specific groups [3], while polarization refers to the exclusive separation of populations within social mobs based on factors such as economics, politics, and religion [4]. These issues lead to social disturbances and conflicts, necessitating effective facilitation to mediate relations between diverse groups within society [5, 6]. Social isolation and fragmentation fundamentally stem from issues related to communication. Previous studies [1, 2] have regarded isolation as a challenge rooted in community dynamics, while others [5, 6] posit fragmentation as a communication issue primarily occurring between in-groups and out-groups.

To tackle these social issues, philosopher Deguchi introduced the concept of the "mixbiotic society," elevating the idea of a symbiotic society to a more advanced level [7, 8]. In a "mixbiotic society," individuals, each possessing unique freedoms and diverse values, mix together in physical proximity with diverse mingling to acknowledge their respective "fundamental incapabilities" and collectively sublimate into solidarity. This concept suggests that the "fundamental incapability" signifies an inherent limitation of an individual, the "I," preventing independent performance of any physical action or the exercise of complete control over others. Consider, for example, the act of riding a bicycle, requiring various elements such as bicycle manufacturing, distribution, road and signal infrastructure, the presence of air, and the force of gravity—all supporting the individual "I." Notably, even a dictator-like "I" cannot exert total control over all others and external factors, highlighting the inability of the human "I" to dominate the entire natural environment.

In the context of the mixbiotic society, the focal point shifts from the "Self as I" to the "Self as WE," where individuals entrust themselves to one another. Achieving this "Self as WE" requires a delicate balance of both openness (freedom) to prevent fragmentation and fellowship (solidarity) from preventing isolation. According to Deguchi, social isolation and fragmentation are viewed as problems of community impoverishment (individual atomization: atomism) and community hypertrophy (in-group crowding: mobism), respectively [9, 10]. Furthermore, Deguchi's conception of "well-going" represents a dynamic way of life, distinct from the static existence of well-being.

Toward a mixbiotic society, we first consider conventional social structure and communication measures. As shown in the following Literature Review section, social network analysis [11, 12], which focuses on static structures, and evolving/temporal network analysis [13, 14], which focuses on structural changes, are well-known methods. Additionally, perspectives emphasizing livingness, such as complexity classification and edges of chaos in cellular automata (CA) [15, 16] and particle reaction-diffusion (PRD) [17], are instructive. These approaches measure dynamically changing patterns of cells and particles based on entropy.

Building on these previous studies, this research aims to develop a novel measure to dynamically evaluate communication patterns within a mixbiotic society, where individual freedom and community solidarity coexist harmoniously. To achieve our goal, we introduce several measures that represent communication patterns as multidimensional vectors. These measures are easier to compute and interpret for dynamic patterns compared with conventional network analysis and entropy-based measures. Next, we hypothesize four-phase mixbiotic society

measures for "atomism" representing isolation, "mobism" representing fragmentation, "mixism" representing a well-going livingness of a mixbiotic society between atomism and mobism, and "nihilism" representing nearly negligible communication. Then, the validity of these new measures is assessed using several representative communication datasets.

The proposed measure embodies a perspective deeply rooted in livingness and well-going, within the context of real society dynamics. This feature sets it apart from both static social network analysis [11, 12] and evolving/temporal network analysis [13, 14], which predominantly focus on aspects such as disease transmission and information diffusion. Additionally, in contrast to the multiscale entropy method [18, 19], a technique that calculates sample entropy across various timescales for time series data, our measure offers the advantage of simplicity in computing the amount and variance of communication patterns, represented as multidimensional vectors. Moreover, our measure is designed according to the irregularity of communication networks in real-world societies, providing greater applicability compared to the phase classification of CA and PRD [17, 20, 21], which are specifically custom-tailored for regular lattice networks.

The remainder of this paper is organized as follows: The following Literature Review section mentions our approach, introducing the literature related to mixbiotic society and the literature on social network analysis, evolving/temporal network analysis, CA, and PRD. The Methods section presents the calculation of the hypotheses for mixbiotic society measures. The Results section presents the evaluation results obtained using the new measures for seven representative real-society datasets. The Discussion section delves into the validity of the new measures, addressing potential issues and discussing their utility compared to conventional network analysis. Finally, the Conclusions section summarizes our findings, provides concluding remarks, and presents avenues for future research.

## Literature review

First, we mention the literature related to Deguchi's mixbiotic society. Anthropologist Graeber and philosopher Karatani proposed a vision of a society where the coexistence of individual freedom and community solidarity is not contradictory, similar to the mixbiotic society. Graeber highlights a "baseline communism" based on the core principles of decentralization, equality, voluntary association, and mutual aid [22, 23]. Similarly, Karatani presents an exchange economic system where individual freedom and community mutual aid coexist, eliminating the negative aspects often associated with the obligation to return in a gift economy [24, 25]. From a community and communication standpoint, in a mixbiotic society where freedom and solidarity harmonize, as illustrated by Deguchi, Graeber, and Karatani, the objective is to transcend the dichotomy and aim for a middle ground between "atomism" and "mobism," which we term a well-going "mixism."

From another perspective, sociologist Luhmann defines a social system as an autopoietic system that forms an emergent order through a network of communication processes involving the generation and disappearance of information [26]. Autopoiesis is a theory of living systems proposed by biologists Maturana and Varela; this theory refers to a self-organizing system consisting of a recursive network of processes in which existing components generate new components through interaction [27]. From the perspective of autopoiesis, to achieve a living social system in which freedom and solidarity coexist harmoniously, it is necessary to evaluate the communication network dynamically, and to take proactive measures to counteract atomism and mobism, guiding society toward a state of well-going mixism.

Next, we discuss the literature related to methods for assessing social structures and communication networks. Social network analysis is a well-known method for assessing the static

structure of social relationships within social systems [11, 12]. In this approach, a network is visually represented as a graph comprising nodes or vertices (individual actors, people, or things) and links or edges (relationships or interactions) [28]. This graph is analyzed using node- and network-level structural measures, including degree, centrality, betweenness, density, cluster coefficient, and modularity. This method has been used in various fields such as sociology, economics, anthropology, and psychology. Recently, it has been applied to assess the structure of food-sharing networks to gauge the risk of starvation [29] and to examine the structure of online emergency collaborative networks during disaster occurrences [30].

To capture a more dynamic perspective, evolving and temporal network analysis techniques are employed [12, 13]. Evolving network analysis examines changes in structural measures to capture network growth and evolution [14]. Temporal network analysis leverages contact sequence graphs and interval graphs represented by nodes and links in a chronological order to observe disease transmission and information diffusion [13]. Recently, these analysis methods have been applied to the study of the evolution of political communities in social media networks [31], and the Susceptible-Infectious-Recovered model has been implemented in temporal networks for understanding infectious epidemics [32].

According to network scientists Holme and Saramäki, the challenges of temporal network analysis include (i) establishing a temporal network generation model that corresponds to the real world, (ii) elucidating the driving mechanism (why contact occurs), and (iii) developing measures of the temporal network structure that simply describe properties [13]. These challenges arise because existing methods are predominantly based and expanded on static structural measures of social network analysis. Recently, to address these challenges, a multiscale entropy (MSE) method has been applied to detect time correlations in temporal networks, and temporal network models have been developed to generate interaction patterns [18]. The MSE method coarsely granulates time series data at various time scales and calculates the sample entropy at each time scale [19]. In the literature [18], multiscale entropy is computed for time series data on the number of activated links and newly activated links in the overall network.

In light of this need for a more dynamic perspective, we temporarily shift our attention away from evolving/temporal network analysis and instead turn to cellular automata (CA). These are mathematical models that are designed to emulate living phenomena. In CA, multitudes of cells are arranged in a spatial lattice, and their pattern evolve over time based on specific update rules that govern their interactions with neighboring cells [15]. Wolfram reported four classes of CA behavior [16]: class 1 is a uniform state, class 2 is a periodic state, class 3 is a random and chaotic state, and class 4 is a mixture of order and randomness.

Computer scientist Langton coined the term "the edge of chaos" to describe class 4. He also introduced a measure of complexity called the $\lambda$ parameter and showed that complexity increases from class 1 to 2, reaches a maximum in class 4, and then decreases in class 3 [33]. Theoretical biologist Kauffman hypothesized that life exists between order and chaos, or on "the edge of chaos" [34]. Theoretical biologist Gunji showed that the "the edge of chaos" concept can be expanded by adjusting the passivity and activity as well as synchrony and asynchrony of the cell update rules [20]. The magnitude of the entropy of cell state and variance in change of the entropy can be measures of "the edge of chaos," although they are not as clear as the $\lambda$ parameter [21].

When viewed differently, the behavior of CA can be understood as a pattern of node states being generated and disappearing as nodes interact within a lattice network. In a similar vein, information scientist Miyakoshi has demonstrated that, analogous to CA, four distinct classes emerge within a mathematical model of particle reaction-diffusion (PRD). This model involves the propagation of two types of particles between the nodes in a two-dimensional lattice network, driven by reaction-diffusion equations [17]. These four classes are systematically charted

on a phase diagram defined by two axes: particle propagation speed and quantity. This mapping is established through a decision formula, which takes into account the mean and range of the mutual information computed over time, as well as the fluctuation component of the standard deviation of particle quantity. The fluctuation component is linked to the variance in the change of entropy, as posited by Gunji's work. Within class 4, the patterns formed by the two types of particles exhibit a life-like behavior between ordered and random patterns.

Drawing inspiration from entropy-based evaluation methods for MSE's temporal interaction patterns, CA's edges of chaos, and PRD's life-like behavior, we find it necessary to evaluate the balance between order and randomness to assess the dynamic livingness between freedom and solidarity in mixbiotic and autopoietic societies. However, because entropy-based methods require computing probabilities for individual events on a defined timescale, a simpler and more real-time method for evaluating order and randomness is desirable. Therefore, our approach for evaluating the dynamic communication patterns revolves around several key parameters. Specifically, we should consider communication patterns as multidimensional vectors and focus on the amount of pattern change, mutual amount before and after the change, and their variance or standard deviation. Considering that the magnitude and variance of entropy change and amount and fluctuation of mutual information represent livingness, we can hypothesize that the mean and variance in the time-varying amount of communication patterns is a valuable measure for characterizing a mixbiotic society.

## Methods

The mixbiotic society measures require a perspective on the livingness of autopoiesis, which implies a well-going mixism between atomism (social isolation) and mobism (social fragmentation), and it is promising to calculate some time-varying amount or its variance for the dynamic patterns of communication. However, a real society is not a regular network or interaction but a complex network or irregular interaction; moreover, it does not always provide sufficient data. Therefore, rather than introducing complex decision formulae for specific mathematical models, it is preferable to use a measure that is easy to compute and interpret, as stated by Holme and Saramäki [13].

Based on these considerations, as a hypothesis regarding mixbiotic society measures, communication patterns in networks formed through communication are regarded as multidimensional vectors. Specifically, let $Q(t)$ be an $n$-dimensional vector of communication patterns at time $t$ on a network consisting of $n$ vertices, and let the element $q_i(t)$ of $Q(t)$ be the information amount that each vertex $i$ ($= 1, 2, \cdots, n$) has at a discrete time $t$, as shown in Eq (1).

$$Q(t) = (q_1(t),\ q_2(t),\ \cdots,\ q_n(t)) \tag{1}$$

We focus on the change between $Q(t)$ and $Q(t + 1)$ to assess livingness and dynamics and then consider the total amount of information (sum of elements), Euclidean distance [35], and cosine similarity [36] as measures that are relatively easy to calculate and whose meaning is easy to interpret. These measures are common in document and image analysis. Although they are sometimes used in network analysis, Euclidean distance is mainly used for the spatial representation of network history (e.g., [37]) and cosine similarity for the clustering of network structure and community detection (e.g., [38]). Cosine similarity is the cosine of the angle between two vectors and is a value in the interval [0, 1] if the components of the vectors are non-negative. Therefore, two vectors in the same direction have a similarity of 1 and two orthogonal vectors have a similarity of 0.

The total amount of information is computed as the sum of the elements $q_i(t)$ within the $n$-dimensional vector $Q(t)$ The change, denoted as $I(t + 1)$, between $Q(t)$ and $Q(t + 1)$ is

calculated using Eq (2). The denominator is for normalization, and we consider the case in which all $n$ vertices have information unit $u$ as the reference. This measure evaluates how the information amount fluctuates through communication.

$$I(t+1) = \frac{\left| \sum_{i=1}^{n} q_i(t+1) - \sum_{i=1}^{n} q_i(t) \right|}{n \cdot u} \tag{2}$$

The Euclidean distance $L(t+1)$ is the distance between $\boldsymbol{Q}(t)$ and $\boldsymbol{Q}(t+1)$, calculated from Eq (3). The denominator is the magnitude of the $n$-dimensional unit vector for normalization. The relative change in the Euclidean distance, $L_R(t+1)$, is calculated using Eq (4). The denominator is the magnitude of the vector of $\boldsymbol{Q}(t+1)$. These measures examine how the communication states separate relatively.

$$L(t+1) = \frac{\sqrt{\sum_{i=1}^{n} (q_i(t+1) - q_i(t))^2}}{\sqrt{n} \cdot u} \tag{3}$$

$$L_R(t+1) = \frac{\sqrt{\sum_{i=1}^{n} (q_i(t+1) - q_i(t))^2}}{\sqrt{\sum_{i=1}^{n} q_i(t+1)^2}} \tag{4}$$

The cosine similarity $S(t+1)$ is the similarity between $\boldsymbol{Q}(t)$ and $\boldsymbol{Q}(t+1)$, calculated using Eq (5). Cosine similarity generally takes a value between −1 and 1, but here it is normalized to a value between 0 and 1 because $q_i(t+1) \cdot q_i(t) \geq 0$. This measure examines the similarity between the communication states. The cosine similarity $S(t+1)$ is a measure of the direction (angle) of the two vectors and is complementary to the Euclidean distance $L(t+1)$. The cosine similarity is also closely related to the correlation coefficient [36].

$$S(t+1) = \frac{\sum_{i=1}^{n} q_i(t+1) \cdot q_i(t)}{\sqrt{\sum_{i=1}^{n} q_i(t+1)^2} \cdot \sqrt{\sum_{i=1}^{n} q_i(t)^2}} \tag{5}$$

Here, the mean and variance of the measures shown in Eqs (2)–(5) are calculated to evaluate the overall trend of communication, assuming that the communication measurement period is from $t = 0$ to $t_{max}$. The mean $\mu_I$ and variance $\sigma_I^2$ of the total information change $I(t)$, mean $\mu_L$ and variance $\sigma_L^2$ of the Euclidean distance $L(t)$, mean $\mu_{LR}$ and variance $\sigma_{LR}^2$ of its relative change $L_R(t)$, and mean $\mu_S$ and variance $\sigma_S^2$ of the cosine similarity $S(t)$ are calculated using Eqs (6)–(9), as well as general calculation formulae. The unbiased variance formula was used to calculate the variance, as shown in the term $t_{max} - 1$ of Eqs (6)–(9).

$$\mu_I = \sum_{t=1}^{t_{max}} I(t)$$
$$\sigma_I^2 = \frac{1}{t_{max} - 1} \sum_{t=1}^{t_{max}} (I(t) - \mu_I)^2 \tag{6}$$

$$\mu_L = \sum_{t=1}^{t_{max}} L(t)$$
$$\sigma_L^2 = \frac{1}{t_{max} - 1} \sum_{t=1}^{t_{max}} (L(t) - \mu_L)^2 \tag{7}$$

$$\mu_{LR} = \sum_{t=1}^{t_{max}} L_R(t)$$

$$\sigma_{LR}^2 = \frac{1}{t_{max} - 1} \sum_{t=1}^{t_{max}} \left( L_R(t) - \mu_{LR} \right)^2$$

(8)

$$\mu_S = \sum_{t=1}^{t_{max}} S(t)$$

$$\sigma_S^2 = \frac{1}{t_{max} - 1} \sum_{t=1}^{t_{max}} \left( S(t) - \mu_S \right)^2$$

(9)

Based on the above measures, we hypothesize the mixbiotic society measures as shown in Table 1. In social isolation (atomism) in row #4 of Table 1, communication is considered separate and sporadic. The communication patterns may resemble chaos and are expected to change in "disorder." Therefore, we hypothesize that the relative change in Euclidean distance is large and that $\mu_{LR}$ is to be applied as a measure $M_{atom}$ of atomism. In social fragmentation (mobism) in row #2, communication is considered in-group biased and crowded. The communication patterns may resemble specific flashing patterns, and they are expected to vary widely in "order." Therefore, we hypothesize that the Euclidean distance is large and that $\mu_L$ is to be applied as a measure $M_{mob}$ of mobism. In mixism between atomism and mobism in row #3, communication is considered to present well-going livingness without becoming sporadic or in-group biased. Communication patterns may resemble the edges of chaos and are expected to vary and circulate between "order" and "disorder." Therefore, we hypothesize that the similarity and dissimilarity balance and that $\mu_S \cdot \sigma_S^2$ is to be applied as a measure $M_{mix}$ of mixism. Note that in a state where there is little or no communication, i.e., nihilism in row #1, the values of $M_{atom}$, $M_{mix}$, and $M_{mob}$ are all considered close to zero.

Here, we show in advance the method for calculating the polar coordinates to visualize the temporal changes in communication pattern $Q(t)$ in the subsequent section. The moving radius $r$ is calculated from the elements of $Q(t)$ as shown in Eq (10), and the declination angle $\theta$ is calculated from the angle between $Q(t)$ and the $n$-dimensional unit vector $\mathbf{1}$ as shown in Eq (11). Note that because $q_i(t) \cdot 1 \geq 0$, $\theta$ is in the range $0 \leq \theta \leq 2/\pi$ (the first quadrant of the two-dimensional coordinate plane).

$$r = \sqrt{\sum_{i=1}^{n} q_i(t)^2}$$

(10)

$$\theta = \text{Arccos} \left( \frac{\sum_{i=1}^{n} q_i(t) \cdot 1}{\sqrt{\sum_{i=1}^{n} q_i(t)^2} \cdot \sqrt{\sum_{i=1}^{n} 1^2}} \right)$$

(11)

Table 1. Hypotheses for mixbiotic society measures.

| # | Phase | Formula | Feature |
|---|---|---|---|
| 1 | Nihilism | $M_{atom}, M_{mix}, M_{mob} \approx 0$ | Static and silent |
| 2 | Mobism (Fragmentation) | $M_{mob} = \mu_L$ | Biased and crowded |
| 3 | Mixism | $M_{mix} = \mu_S \cdot \sigma_S^2$ | Balanced and cyclic |
| 4 | Atomism (Isolation) | $M_{atom} = \mu_{LR}$ | Separate and sporadic |

Now, in validating the mixbiotic society measures, we used seven temporal network datasets with potentially different communication pattern features. These datasets include the network data repository collected by Rossi and Ahmed [39] and face-to-face contact datasets collected jointly by the ISI Foundation, the French National Centre for Scientific Research (CNRS, *Centre national de la recherche scientifique*), and Bitmanufactory [40]. Note that contact data are considered to approximate face-to-face communication [41]. Co-location or co-presence is related to communication [42, 43], and co-presence data with high spatial resolution corresponds to face-to-face communication data [44]. Incidentally, the communication distance of the RFID sensor or radio badge used for contact detection was less than 1.5 m, compared to approximately 10 m for Bluetooth.

Of the seven datasets used, the first comprised contact data between students in a French high school, measured using RFID sensors [45, 46]. The second dataset comprised contact data between children and teachers in a French primary school [47, 48]. The third comprised contact data between individuals at the workplace in a French institute office [49, 50]. The fourth comprised contact data between individuals including adults, adolescents, and children in a rural village in Malawi [51, 52]. The fifth comprised data between participants measured using radio badges at the Hypertext 2009 conference at Torino [53, 54]. The sixth comprised data on messages sent and received in the online community (Facebook) for students at the University of California, Irvine [55, 56]. Finally, the seventh comprised data on emails sent and received by the U.S. Democratic National Committee [57, 58].

These datasets were selected because in high school and primary school, the same members often move in groups, making their communication patterns similar to mobism. In the workplace and village, real-life communication occurs, resembling a mixism that balances mobism and atomism. Conferences involve special communication, but they may be relatively close to mixism owing to the dynamics of meeting and parting. Online communities and email involve sporadic communication in virtual spaces, making them closer to atomism. Therefore, we believe that these datasets are sufficient to validate the mixbiotic society measures, because they correspond to mobism, mixism, and atomism, except for nihilism, which is the absence of communication.

## Results

Before evaluating the seven datasets by the mixbiotic society measures, a social network analysis of these datasets was performed. Table 2 lists the graph features of the network for all times in the datasets. The symbol $\infty$ in Table 2 indicates that some vertices were not connected throughout the time. $t_{count}$ is the number of data points at the time recorded in each dataset, and $t_{max}$ is the number of data points counted at the same time as 1, similar to how $t_{max}$ is used

**Table 2. Network graph features of seven datasets.**

|  | High school [45, 46] | Primary school [47, 48] | Workplace [49, 50] | Village [51, 52] | Conference [53, 54] | Online community [55, 56] | Email [57, 58] |
|---|---|---|---|---|---|---|---|
| Vertex count | 327 | 242 | 217 | 86 | 113 | 1,899 | 1,891 |
| Edge count | 5,818 | 8,317 | 4,274 | 347 | 2,498 | 22,195 | 5,598 |
| Diameter | 4 | 3 | 5 | $\infty$ | 3 | $\infty$ | $\infty$ |
| Mean distance | 2.159 | 1.732 | 1.882 | $\infty$ | 1.656 | $\infty$ | $\infty$ |
| Density | 0.1092 | 0.2852 | 0.1824 | 0.0949 | 0.3470 | 0.0077 | 0.0025 |
| Mean cluster coefficient | 0.504 | 0.526 | 0.381 | 0.527 | 0.535 | 0.109 | 0.209 |
| $t_{count}$ | 188,508 | 125,773 | 78,439 | 102,293 | 20,818 | 61,734 | 39,264 |
| $t_{max}$ | 7,375 | 3,100 | 18,488 | 43,438 | 5,246 | 60,774 | 21,751 |

in Eqs (6)–(9). Fig 1 shows the radar chart of graph features. For ease of viewing, the values for each item were normalized using the maximum value for every model. The ratio $t_{max}/t_{count}$, which is close to 1 in Fig 1, indicates that co-temporal communication was negligible. As shown in Fig 1, the online community and email were characterized by smaller graph density and mean cluster coefficients and larger $t_{max}/t_{count}$ than the other five cases, while no other obvious feature differences were found over the seven cases. Note that the network graphs, except for that of the village, can be found in [45, 47, 49, 53, 55, 57].

Table 3 and Fig 2 present the results of calculating the mixbiotic society measures according to the method described in Eqs (1)–(9). Note that in the seven datasets, the duration of the communication and number of persons involved therein were recorded but not the content or amount of information. Therefore, the value of the element $q_i(t)$ of $\mathbf{Q}(t)$ in Eq (1) was calculated as 0 or 1. In Fig 2A, for ease of understanding, we present a radar chart with seven cases: the high school and primary school (dark and light green), workplace, village and conference (dark, medium and light orange), and online community and email (dark and light blue). For ease of viewing, the values for each item were normalized using the maximum value for every model. In Fig 2B, for ease of comparison, the primary school and conference are omitted from the seven cases.

As shown in Fig 2, the high school and elementary school have larger $\mu_L$ than those of the other five cases. Their similarity $\mu_S$ is larger, but their variances $\sigma_S^2$ and $\mu_S \cdot \sigma_S^2$ are smaller. This is because in both schools, relatively similar members always gather in the same place to communicate, causing similar communication patterns. Note that $\mu_I$, $\sigma_I^2$ and $\sigma_L^2$ are larger in the elementary school than in the high school, inferring that the elementary school has more meeting and parting of the same members (and thus more variation in the information

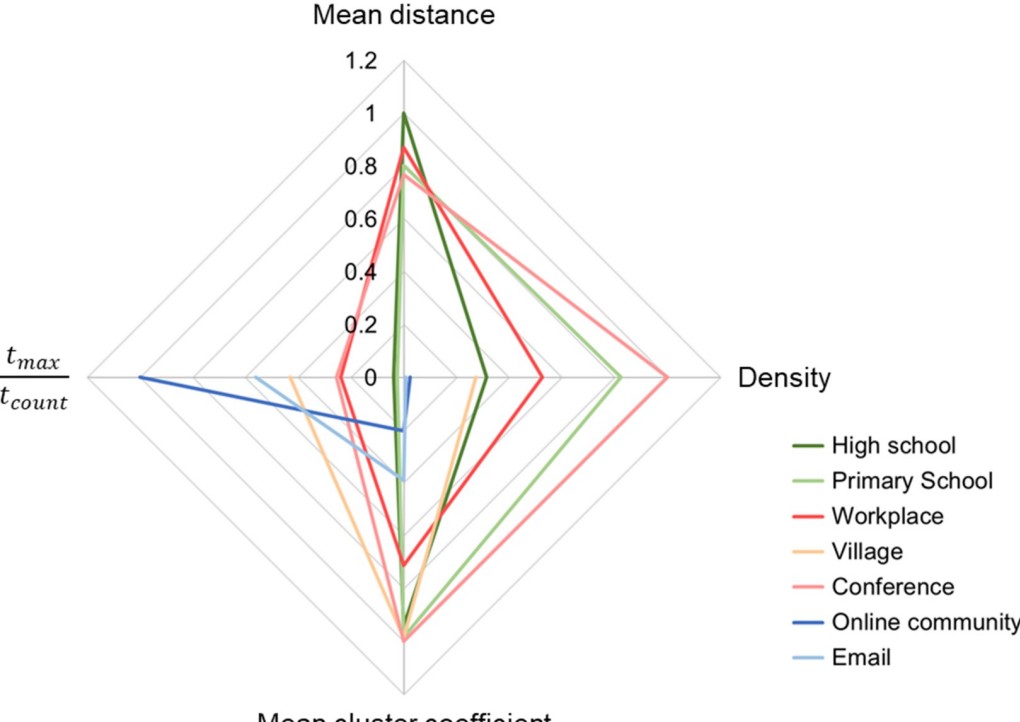

**Fig 1. Radar chart of graph features based on social network analysis.** Seven cases: high school, primary school, workplace, village, conference, online community, and email.

**Table 3. Calculation results on seven datasets for mixbiotic society measures.**

|  | High school | Primary school | Workplace | Village | Conference | Online community | Email |
|---|---|---|---|---|---|---|---|
| $\mu_I$ | 0.0213 | 0.0546 | 0.0118 | 0.0176 | 0.0234 | 0.0000 | 0.0014 |
| $\sigma_I^2$ | 0.0004 | 0.0021 | 0.0002 | 0.0004 | 0.0009 | 0.0000 | 0.0000 |
| $\mu_L$ ($M_{mob}$) | 0.3266 | 0.6673 | 0.1560 | 0.1506 | 0.2171 | 0.0411 | 0.0750 |
| $\sigma_L^2$ | 0.0133 | 0.0254 | 0.0091 | 0.0130 | 0.0235 | 0.0002 | 0.0109 |
| $\mu_{LR}$ ($M_{atom}$) | 0.7394 | 0.9128 | 0.8151 | 0.6463 | 0.7954 | 1.2431 | 1.6112 |
| $\sigma_{LR}^2$ | 0.0253 | 0.0236 | 0.2070 | 0.2652 | 0.2648 | 0.1716 | 4.1790 |
| $\mu_S$ | 0.7287 | 0.5933 | 0.6439 | 0.7158 | 0.6431 | 0.1594 | 0.1573 |
| $\sigma_S^2$ | 0.0108 | 0.0114 | 0.0722 | 0.0864 | 0.0860 | 0.0931 | 0.0636 |
| $\mu_S \cdot \sigma_S^2$ ($M_{mix}$) | 0.0078 | 0.0067 | 0.0465 | 0.0619 | 0.0553 | 0.0148 | 0.0100 |

amount $I(t)$ and distance $L(t)$). From the above, $\mu_L$ is considered useful as a mobism measure $M_{mob}$.

The workplace, village and conference have both larger similarity $\mu_S$ and its variance $\sigma_S^2$ than those of the other four cases, and therefore $\mu_S \cdot \sigma_S^2$ is larger. This is because the same members gathered according to the time of day and then gathered with other members when the time of day changed (see [52] for the tendency of temporal activity in the village). Note that $\sigma_L^2$ is larger in the conference than in the workplace and village, inferring the peculiarity of conference, which differs from daily work and life. From the above, the composite indicator $\mu_S \cdot \sigma_S^2$ is useful as a mixism measure $M_{mix}$ that balances between social isolation (atomism) and fragmentation (mobism), i.e., between similarity and dissimilarity.

Online community and email have a peculiar shape compared to the other five cases. Both have larger $\mu_{LR}$, and their dissimilarity $\sigma_S^2$ is larger, but their similarity $\mu_S$ and $\mu_S \cdot \sigma_S^2$ are smaller. This is because the instantaneous change in communication patterns is large, that is, the communication is separate and sporadic. Note that email has a larger $\sigma_{LR}^2$ than online community, presumably because email has an even smaller temporal context of communication. From the above, $\mu_{LR}$ is considered useful as an atomistic measure $M_{atom}$.

In addition, comparing Fig 2 to Fig 1A, Fig 1A distinguishes the charts of the seven cases better than Fig 2. Mixbiotic society measures better represent the characteristics of the communities than social network analysis.

Fig 3 shows the communication trajectories drawn on the polar coordinates according to the method described in Eqs (10) and (11). The trajectory was drawn by connecting the polar coordinates of $\mathbf{Q}(t)$ at each time $t$ with a line segment in sequence in the period from $t = 0$ to $t_{max}$. Fig 3A–3F show the high school, primary school, workplace, village, online community, and email, respectively.

Although difficult to see owing to the dense overlap of trajectories, the high school in Fig 3A and elementary school in Fig 3B show the aspect of a large reciprocal movement from the lower left to upper right of the coordinates due to mobism-like collective action. Compared to these, the workplace in Fig 3C and village in Fig 3D have a shorter length of trajectories (i.e., less change and more similarity) but a wider declination angle (i.e., combined dissimilarity), giving them the appearance of a mixism.

The online community in Fig 3E and e-mail in Fig 3F occasionally show trajectories that extend to the upper right of the coordinates owing to events such as simultaneous

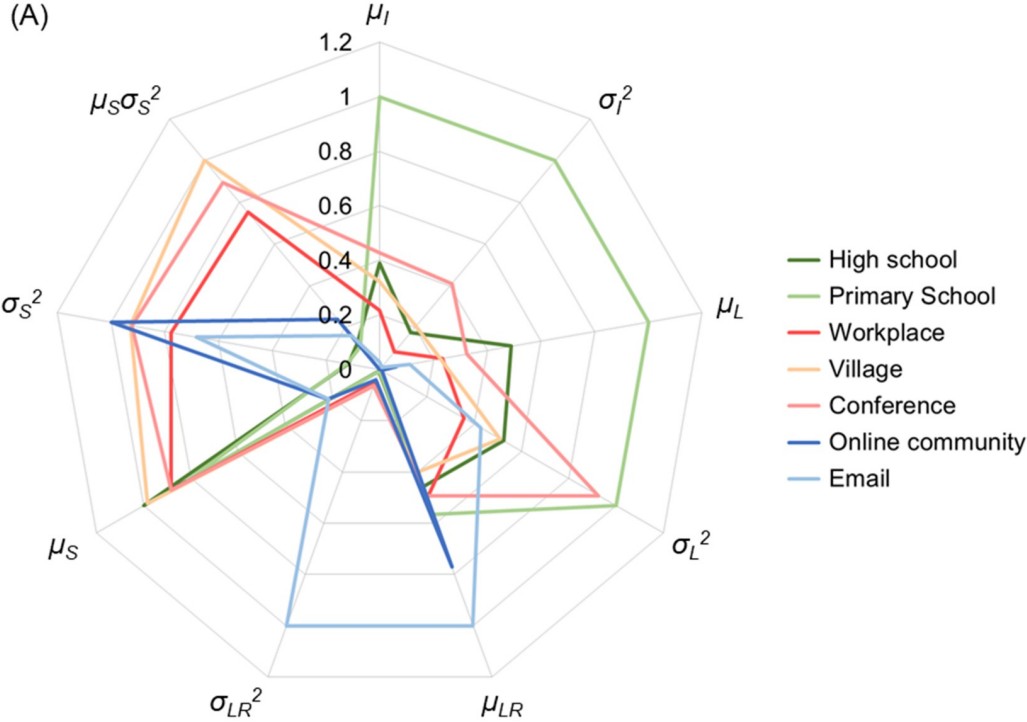

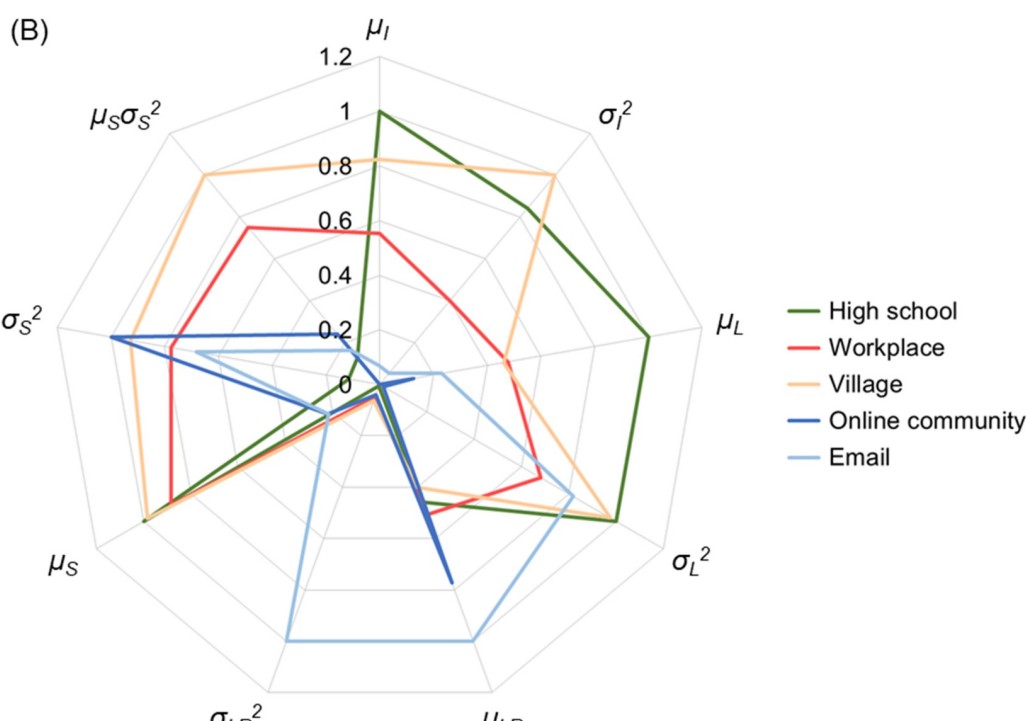

**Fig 2. Radar chart of calculation results for mixbiotic society measures.** (A) Seven cases: high school, primary school, workplace, village, conference, online community, and email. (B) Five cases: high school, workplace, village, online community, and email.

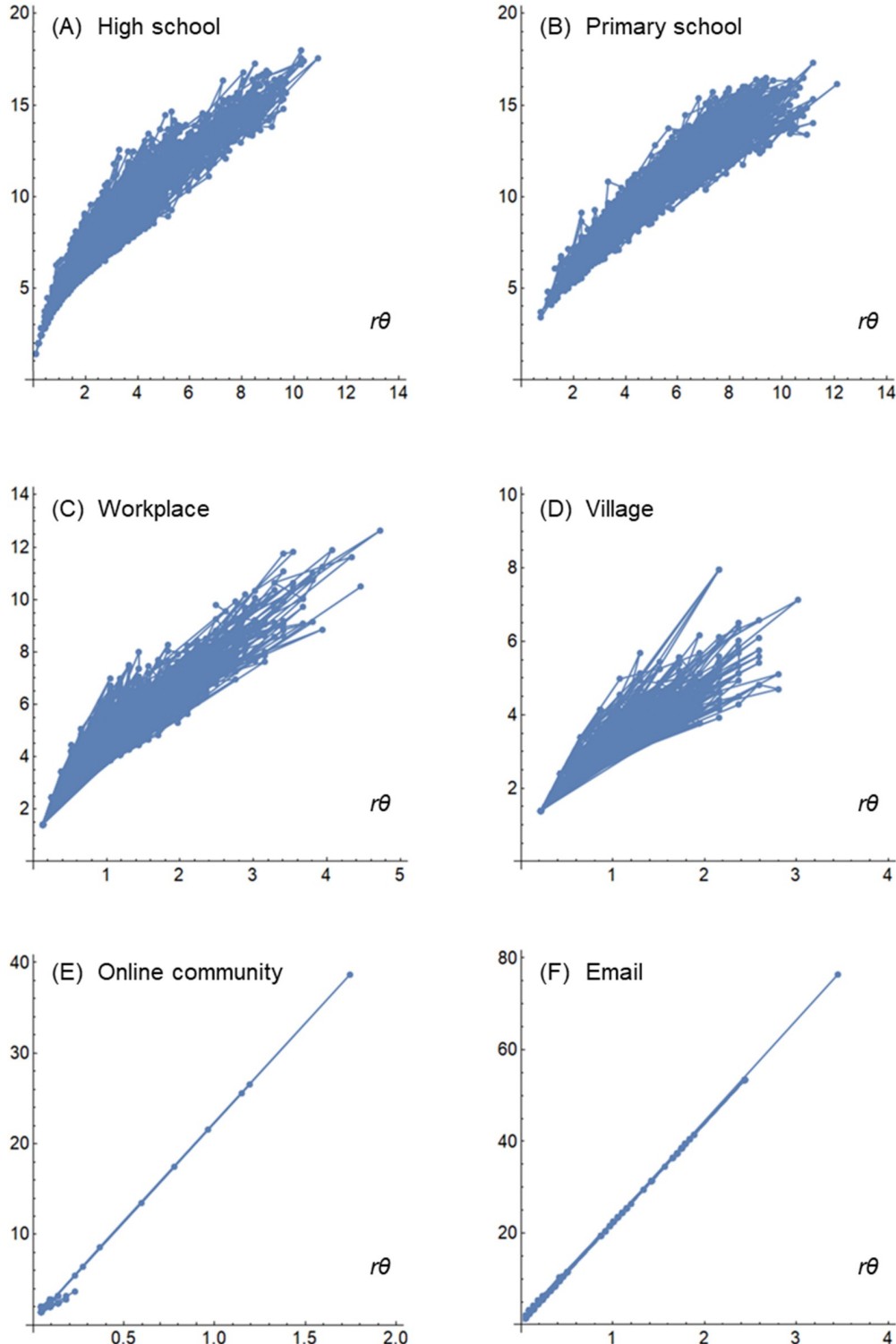

**Fig 3. Communication trajectories.** (A) High school, (B) primary school, (C) workplace, (D) village, (E) online community, and (F) email.

transmissions, but there are locally sporadic and repetitive trajectories that overlap densely in the lower left of the coordinates, giving the appearance of an atomism.

The findings from the calculations in Table 3, radar charts in Fig 2, and trajectories in Fig 3 with respect to the real-society datasets, validate the hypotheses for mixbiotic society measures shown in Table 1. However, as discussed in subsequent sections, further validation of mixbiotic society measures using various datasets is needed, such as datasets on social isolation of elderly in a local community or on social fragmentation among groups with polarized political opinions.

On a different note, the study in [18] presents a multiscale entropy analysis on elementary school, high school, workplace, hospital, and conference datasets from the same dataset collection [40] that we used. According to this analysis, there are three categories in terms of time correlation of sample entropy: elementary school, workplace and hospital, and high school and conference. These differ from the categories in the mixbiotic society measures: elementary school and high school, and workplace and conference. Although we have not yet examined the reasons in detail, because multiscale entropy analysis and mixbiotic society measures have different perspectives of analysis, they could be used in a complementary manner.

## Discussion

By capturing temporal changes in communication patterns in the network, as in this study, rather than temporal changes in network structure as in conventional temporal network analysis, it is possible to identify communication features and typify the community. In real communities such as workplaces, villages, and conferences, as opposed to special environments such as a high school and primary school and virtual communities such as online community and email, the fact that the measure of mixism $M_{mix} = \mu_S \cdot \sigma_S^2$, balancing similarity and dissimilarity, is large provides several suggestions.

Similarity is the "mixing" of humans with physical proximity, whereas dissimilarity is the "mingling" of diverse humans. These points indicate what philosopher Deguchi calls a "mixbiotic society" [7]. Mixism indicates a balance between the similarity and dissimilarity of communication, that is, the community is in what Deguchi refers to as a dynamic and well-going state of living as opposed to a static state of being. The clear difference between real and virtual communities based on the presented measures reminds us of the broad importance of physical proximity and face-to-face contact. Communication through bidirectional, multimodal, and face-to-face contact is more important and preferable for mixbiotic and well-going communities than one-way, transient virtual communication.

The balance between passivity and activity, as well as synchrony and asynchrony according to findings based on CA [20, 21] and PRD [17] are key to livingness (the edge of chaos). These findings inspire the importance of a balance between habituality and uncertainty, as well as that of similarity and dissimilarity, in communication in mixbiotic societies. This is consistent with the recognition that both bonding and bridging types of social capital are important (e.g. [59]). In the future, measuring mixism, $M_{mix} = \mu_S \cdot \sigma_S^2$, for various real communities will aid in clarifying the features and improvement policies of these communities. For instance, it can foster the establishment of communal spaces for local residents and facilitate assistance for exchange activities [60]. Additionally, it can play a pivotal role in transitioning corporate structures from hierarchical to Teal organizations [61].

Regarding virtual communities, we did not evaluate social networking services (SNSs), characterized by short messages and prevalent images, in this study, but it is easy to imagine that the trend toward atomism will become even stronger in light of the results for online communities and emails. Simultaneously, we can also predict the emergence of mobistic

phenomena such as those seen in SNS flare-ups. Although virtual communication is not expected to completely replace face-to-face communication, its soundness depends on developments such as the implementation of bidirectional and continuous, as opposed to unidirectional or transient, communication technology supporting such multimodal communication and its affordances, as well as services combining face-to-face communication.

In this study, for the online community and emails, we used data related to the time of sending and receiving messages; however, replies to the same message can be considered continuation of communication and accumulation of information over time. If we could collect data on replies in addition to sending and receiving times, we might be able to obtain different calculation results for the measures and trajectories. The same can be said for replies and thread continuation on SNS. Although online meetings and multiplayer video chats are considered similar to real communication, the issue is that a substantive range of communication time must be measured because, in contrast to place-sharing in face-to-face contact, there are cases in which a meeting or chat screen is open without communication. In addition, if the dynamic pattern of communication with avatars and AI in a virtual space is similar to that of real communication, whether it is desirable as a mixbiotic society cannot be determined only by the measures set here; we will need to combine it with other psychological measures and ethnomethodology. For example, when avatars (digital representations of individuals) engage in communication, they may have a different psychological impact than face-to-face contact.

## Conclusions

This section summarizes our contributions, provides concluding remarks, and findings and presents avenues for future challenges and developments. In summary, to address the social issues of isolation and fragmentation, developing methods that assess dynamic communication alongside static social relations is essential. As a contribution to sociological and network science disciplines, we proposed new measures for mixbiotic societies, which can reveal shifts in communication patterns. Specifically, we first hypothesized the mixbiotic society measures by considering communication patterns as multidimensional vectors, corresponding to the four phases of mixism (balanced and cyclic), atomism (separate and sporadic), mobism (biased and crowded), and nihilism (static and silent). Next, through validation with seven real-society datasets, we demonstrated the usefulness of the composite measure of mixism, $M_{mix} = \mu_S \cdot \sigma_S^2$, and the possibility of typifying communities based on the plural measures.

In conclusion, the mixbiotic society measures are superior to conventional social network analysis in that they can easily calculate temporal changes in communication on a network; therefore, the meaning of measures is easy to interpret. We believe that the mixism measure can be utilized as a measure of balanced well-going between similarity and dissimilarity, i.e., between proximate mixing and diverse mingling toward a desirable mixbiotic society. Combining the present measures with the structural analysis of social network and multiscale entropy analysis of temporal network will provide more insights. For example, comparing the mixism measure with social network analysis will reveal whether a communication issue stems from frequency and bias or a lack of network paths. Additionally, examining both the mixism measure and multiscale entropy analysis will determine whether the issue is diachronic or dependent on the time of day or day of the week. Identifying issues in this manner should help concretize methods for bridging isolation and fragmentation, such as mediating communication, facilitating consensus, and scheduling across time zones. However, these mixbiotic society measures have a potential disadvantage in that, similar to conventional network analysis, they do not consider the content of communication. Therefore, we suggest the integration of these measures with analyses of language, emotion, and behavior. For example, vectorization

of language in communication could be evaluated similarly to mixbiotic society measures. Such a combined approach holds the potential to deliver deeper insights into the dynamics of mixbiotic societies.

In future research, verifying and reviewing the validity of atomism and mobism measures using datasets with actual atomism and mobism as well as datasets related to SNS and social media will be necessary. Further datasets would allow for correlation analysis with community features and principal component analysis for typologies. The perspectives for setting measures include simplicity of calculation, ease of interpretation of meaning, and ease of comparison among different communities. To enhance precision and resolution by introducing more complex measures, the potential trade-off with versatility must be carefully weighed. Consequently, we aim to approach this issue from a multifaceted perspective. This approach may involve comparing and contrasting the benefits and drawbacks of both highly precise and exceptionally versatile measures. Additionally, we intend to perform correlation and principal component analyses between survey results and these measures to better understand their practical usefulness.

Note that the mixbiotic society measures do not analyze the network structure at each time but rather analyze communication patterns in the network and thus have the limitation of requiring network information over the entire period beforehand, which at first glance does not seem suitable for real-time analysis. In practice, however, it is sufficient to calculate the measures from the start to the present time as needed, adding elements of the multidimensional vector of the information set as time passes from the start. Moreover, due to the manageable computational load, this analysis can be conducted in near-real time. Additionally, while the measures currently quantify time as the number of communication events, it is possible to explore an alternative approach, such as weighting multidimensional vectors based on communication duration.

Future developments require validation using datasets based on empirical research through fieldwork. Mixbiotic society measures can be described as objective measures of communication dynamics and livingness but in combination with subjective measures, such as the Self-as-WE scale [62] and the general well-being scale [63], which are appropriate for a mixbiotic society. To encourage societies and communities to work well, we recommend that they be used comprehensively. As a candidate approaches fieldwork, it would be interesting to target sites where social isolation and fragmentation are issues and sites that span the real and virtual worlds of digital democracy [64] and platform cooperativism [65], where communication is critical. These contexts present unique challenges, including the need to balance diverse discussions and consensus among participants in a digital democracy and to strike a harmony between individual freedom and collective solidarity within platform cooperativism.

## Acknowledgments

The authors received valuable comments from the Hitachi Kyoto University Laboratory of the Kyoto University and the Hitachi, Ltd. The authors express their gratitude to the academic editor and anonymous reviewers, and Editage (www.editage.com) for English proofreading.

## Author Contributions

**Conceptualization:** Takeshi Kato, Jyunichi Miyakoshi, Tadayuki Matsumura, Ryuji Mine, Hiroyuki Mizuno, Yasuo Deguchi.

**Data curation:** Takeshi Kato.

**Formal analysis:** Takeshi Kato.

**Funding acquisition:** Yasuo Deguchi.

**Investigation:** Takeshi Kato, Jyunichi Miyakoshi, Tadayuki Matsumura.

**Methodology:** Takeshi Kato, Jyunichi Miyakoshi, Tadayuki Matsumura.

**Project administration:** Ryuji Mine, Hiroyuki Mizuno, Yasuo Deguchi.

**Software:** Takeshi Kato.

**Supervision:** Ryuji Mine, Hiroyuki Mizuno, Yasuo Deguchi.

**Validation:** Takeshi Kato.

**Visualization:** Takeshi Kato.

**Writing – original draft:** Takeshi Kato.

**Writing – review & editing:** Takeshi Kato, Ryuji Mine, Hiroyuki Mizuno.

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
