## [Decision Letter · Decision Letter 0]

24 May 2024

PONE-D-23-39776Mixbiotic society measures: Assessment of community well-going as living systemPLOS ONE

Dear Dr. Kato,

Thank you for submitting your manuscript to PLOS ONE. After careful consideration, we feel that it has merit but does not fully meet PLOS ONE’s publication criteria as it currently stands. Therefore, we invite you to submit a revised version of the manuscript that addresses the points raised during the review process.

We look forward to receiving your revised manuscript.

Kind regards,

Isaac Akintoyese Oyekola

Academic Editor

PLOS ONE

Additional Editor Comments:

Following reviewers’ comments, the manuscript will further benefit from the following:

(1) Clear indication/presentation of study methodology and results both in the abstract and main text.

(2) Logical consistency

(3) Minor grammatical corrections such as usage of ‘Conversely’ in the fourth sentence of the Introduction instead of ‘On the other hand’, since the two significant issues in contemporary society (Social isolation and fragmentation) both present challenges, as well as avoid starting a sentence with acronym, among others.

(4) Lesser number of references, to include only relevant ones

(5) Presenting study methodology under the ‘Methods’ section, as against describing methodology in the opening sentences of the Result Section. In fact, clear description of Methodology is vital for better understanding and replicability of findings in future studies.

(6) Appropriate caption of the conclusion section to reflect the content of the section, as also seen in the last sentence of the Introduction section.

(7) Complete and accessible reference sources as contained in the URL. That is, some URL were not accessible such as the 8th list of reference.

Generally, the manuscript borrows enormously from many disciplines, thereby demonstrating its multidisciplinary nature. Readers will be interested to know the discipline where such study can be easily situated. That probably makes the manuscript hard to follow easily since it employs varying concepts that may be outside the scope of reader’s understanding. Authors are therefore advised to simplify the texts for common readers to benefit, clearly explaining its relevance, implications and recommendations.

Reviewers' comments:

Reviewer's Responses to Questions

**Comments to the Author**

1. Is the manuscript technically sound, and do the data support the conclusions?

Reviewer #1: Yes

Reviewer #2: Partly

Reviewer #3: Yes

2. Has the statistical analysis been performed appropriately and rigorously? 

Reviewer #1: I Don't Know

Reviewer #2: I Don't Know

Reviewer #3: Yes

3. Have the authors made all data underlying the findings in their manuscript fully available?

Reviewer #1: Yes

Reviewer #2: Yes

Reviewer #3: No

4. Is the manuscript presented in an intelligible fashion and written in standard English?

Reviewer #1: Yes

Reviewer #2: No

Reviewer #3: Yes

5. Review Comments to the Author

Reviewer #1: 1) Throughout the text, different objectives are mentioned (line 30, line 115, line 200, line 211) and none of them clearly describes the objective of the article, which was: to develop, present and apply innovative measures of the mixbiotic society to evaluate dynamic communication patterns , complementing the traditional static analysis of social networks. Furthermore, the objective must be included in the introduction.

2) The justification must also be included in the introduction. It is possible to use the text from lines 213 to 233.

3) On the text (line 306): "To validate the mixbiotic society measures, we performed calculations on seven temporal network datasets with potentially different communication pattern features". The data in this study is enough to validate it?

4) The text from lines 308-327 must appear in methods and also in the abstract.

5) The sample used in the study should be mentioned on the abstract.

Reviewer #2: PLOS ONE

Comments on Manuscript ID PONE-D-23-39776: “Mixbiotic society measures: Assessment of community well-going as living system”

Thank you for inviting me to review this work. In this manuscript, the authors use seven real-society datasets to present mixbiotic society measures to evaluate dynamic communication networks in society. Social network analysis is not my area of expertise, and I assume the journal has experts commenting on the methods of the contents, so I will provide feedback on the clarity and accessibility of the article to the general readership of PLOS ONE. In this spirit, I offer critical feedback below in the hopes of improving the contribution of the manuscript.

Substantive comments:

My biggest concern with this manuscript is that it is hard to follow:

1. For the front end, there are some parts that should be included in the literature review section. For instance,

a. from lines 87 to 108, the authors demonstrate different scholars’ perspectives about the social system. These theoretical backgrounds help readers understand the relationship between individuals and communities. However, these parts should be included in the literature review.

b. The authors didn’t mention nihilism until line 293, but the concept is in the abstract, so it should be introduced earlier in the literature review section to avoid confusion.

c. The authors discuss their datasets starting from line 308, and this should be included in the method section.

Second, the authors use different settings, e.g., high school, primary school, workplace, etc., in their dataset without giving justifications. How do these diverse settings contribute to the overall research objectives? How do the mixbiotic society measures gauge the activities in different settings?

Thirdly, it would be helpful if the authors could explain more about the application of this dynamic network analysis versus the traditional static analysis in the conclusion section. For example, they wrote: “the mixism measure can be utilized as a measure of balanced well-going between similarity and dissimilarity, proximate mixing and diverse mingling toward a desirable mixbiotic society. Combining the present measures with the structural analysis of social network and multiscale entropy analysis of temporal network will provide more insights”. It would be worth expanding the potential implications in real-world settings.

Methodological comments:

Including the temporal changes in the communication network measure is a very interesting idea, but as a non-network analysis person, I found the following area relatively inaccessible:

The authors are not very clear about how introducing the methods of CA and PRD can better measure the dynamic perspective of network analysis in the literature section. What is the connection between these methods? I wish I could see more elaborated explanations in the paragraph starting with line 198.

Reviewer #3: The article is very good and the topic is interesting; I believe that it needs further studies to learn about the application of this type in regions and neighborhoods.

But the clarity of the idea and the information is not precisely focused, and how is it possible to connect isolation, fragmentation, and the mixed neighborhood? I think the writer should expand on that.

6. PLOS authors have the option to publish the peer review history of their article (what does this mean?). If published, this will include your full peer review and any attached files.

Reviewer #1: No

Reviewer #2: No

Reviewer #3: No

---

## [Author Response · Author response to Decision Letter 0]

8 Jun 2024

Responses to the Academic Editor and reviewers are listed in the file labeled 'Response to Reviewers.'

---

## [Editor Report · Decision Letter 1]

4 Jul 2024

Mixbiotic society measures: Assessment of community well-going as living system

PONE-D-23-39776R1

Dear Dr. %Kato%,

We’re pleased to inform you that your manuscript has been judged scientifically suitable for publication and will be formally accepted for publication once it meets all outstanding technical requirements.

Kind regards,

Isaac Akintoyese Oyekola

Academic Editor

PLOS ONE

Additional Editor Comments (optional):

Following careful attention to Editor’s and reviewers’ comments, the manuscript may be considered for publication.

However, authors must address the following:

• Correct the following sentence: “In [18], multiscale entropy is computed for time series data on the number of activated links in the overall network.” Reverting to previous sentence may be better.

• Avoid starting a sentence with acronym.

• The Conclusion section should read, “Summary, Conclusion and Future Research”.
---

## [Editor Report · Acceptance letter]

15 Jul 2024

PONE-D-23-39776R1 

PLOS ONE

Dear Dr. Kato, 

I'm pleased to inform you that your manuscript has been deemed suitable for publication in PLOS ONE. Congratulations! Your manuscript is now being handed over to our production team.

Kind regards, 

on behalf of

Dr. Isaac Akintoyese Oyekola 

Academic Editor

PLOS ONE